# Scalable Rule Learning in Probabilistic Knowledge Bases

**Arcchit Jain**                                              ARCCHIT.JAIN@CS.KULEUVEN.BE
*KU Leuven*

**Tal Friedman**                                                      TAL@CS.UCLA.EDU
*University of California, Los Angeles*

**Ondřej Kuželka**                                        ONDREJ.KUZELKA@CS.KULEUVEN.BE
*KU Leuven*

**Guy Van den Broeck**                                              GUYVDB@CS.UCLA.EDU
*University of California, Los Angeles*

**Luc De Raedt**                                          LUC.DERAEDT@CS.KULEUVEN.BE
*KU Leuven*

## Abstract

Knowledge Bases (KBs) are becoming increasingly large, sparse and probabilistic. These KBs are typically used to perform query inferences and rule mining. But their efficacy is only as high as their completeness. Efficiently utilizing incomplete KBs remains a major challenge as the current KB completion techniques either do not take into account the inherent uncertainty associated with each KB tuple or do not scale to large KBs.

Probabilistic rule learning not only considers the probability of every KB tuple but also tackles the problem of KB completion in an explainable way. For any given probabilistic KB, it learns probabilistic first-order rules from its relations to identify interesting patterns. But, the current probabilistic rule learning techniques perform grounding to do probabilistic inference for evaluation of candidate rules. It does not scale well to large KBs as the time complexity of inference using grounding is exponential over the size of the KB. In this paper, we present SafeLearner– a scalable solution to probabilistic KB completion that performs probabilistic rule learning using lifted probabilistic inference – as faster approach instead of grounding.

We compared SafeLearner to the State-of-the-art probabilistic rule learner ProbFOIL$^+$ and to its deterministic contemporary AMIE+ on standard probabilistic KBs of NELL (Never-Ending Language Learner) and Yago. Our results demonstrate that SafeLearner scales as good as AMIE+ when learning simple rules and is also significantly faster than ProbFOIL$^+$.

## 1. Introduction

There is an increasing tendency to construct knowledge bases and knowledge graphs by machine learning methods. As a result, knowledge bases are often incomplete and also uncertain. To cope with uncertainty, one often resorts to probabilistic databases and logics [Van den Broeck and Suciu, 2017, De Raedt et al., 2016], which take into account the probability of the tuples in the querying process. The most widely used probabilistic database semantics is based on the tuple-independent probabilistic databases model, which assumes that every tuple in every table of the database is independent of one another

To cope with incomplete knowledge bases, various researchers have used machine learning techniques to learn a set of rules that can be used to infer new tuples from the existing ones, thereby completing the knowledge base [Betteridge et al., 2009]. This traditional relational rule learning setting [Quinlan, 1990] has been extended to probabilistic logics and databases by De Raedt et al. [2015]. However, the ProbFOIL approach of De Raedt et al. suffers from one key limitation: It does not scale well to large databases due to the grounding step, which results in an intractable probabilistic inference problem. The key contribution of this paper is the introduction of the SafeLearner system which performs two major tasks. 1) It uses lifted inference to avoid the grounding step and to improve scaling. 2) It enhances a highly efficient rule generation system, AMIE+ [Galárraga et al., 2015] to obtain deterministic candidate rules which are then made probabilistic using lifted inference.

This paper is organized as follows. We introduce the background for this paper in Section 2. We define, in Section 3, the problem of learning a set of probabilistic rules. Sections 4 and 5 outline the idea behind the working of SafeLearner. Section 6 proposes the algorithm for SafeLearner. In Section 7, we present an experimental evaluation in the context of the NELL knowledge base [Betteridge et al., 2009]. An overview of related work can be found in Section 8. Section 9 discusses future research directions and concludes.

## 2. Background

The notation used and definitions introduced in this section are largely adapted from the work of Ceylan et al. [2016]. Throughout the paper, $\sigma$ denotes a relational vocabulary, consisting of a finite set of predicates or relations $\mathcal{R}$, and a finite set of constants or entities $\mathcal{C}$. The *Herbrand Base* of $\sigma$ is the set of all ground atoms, called $\sigma$-*atoms*, of the form $R(c_1, ..., c_n)$ with $R \in \mathcal{R}$ and $c_i \in \mathcal{C}$ (that is, all tuples that can be constructed from $\mathcal{R}$ and $\mathcal{C}$). A $\sigma$-interpretation is a truth value assignment to all the $\sigma$-atoms, often represented as a set containing all the $\sigma$-atoms mapped to *True*. A relational database can be seen as a $\sigma$-interpretation [Abiteboul et al., 1995]. Here, database relations correspond to predicates and database tuples correspond to $\sigma$-atoms. The $\sigma$-atoms that are true in the $\sigma$-interpretation are those that are listed in the database tables.

### 2.1 Queries

In databases (represented as $\sigma$-interpretations), query answering can be formalized as follows. A query is a first-order logic formula $Q(x_1, x_2, \ldots, x_k)$ with free variables $x_1$, $x_2$, $\ldots$, $x_k$. The task of answering the query translates to finding the set of all substitutions $[x_1/t_1, x_2/t_2, \ldots, x_k/t_k]$ such that $\omega \models Q[x_1/t_1, x_2/t_2, \ldots, x_k/t_k]$. The answer to a query $Q$ w.r.t. a database $\omega$ is denoted as $\text{Ans}(Q, \omega)$.

**Example 1** Consider the database consisting of Tables 1, 2 and 3 and the query $Q_1(x) =$ location$(x, \text{mit}) \wedge$ coauthor$(x, \text{fred})$. Then the answer to the query $Q$ consists of the following substitutions: $[x/\text{alice}]$ and $[x/\text{dave}]$. In general, queries may also contain variables bound by existential or universal quantifiers. For instance, if we have the query $Q_2(z) = \exists x, y.$ location$(z, x) \wedge$ coauthor$(z, y)$, which asks for researchers from the database

| researcher | paper | P |
|:---:|:---:|:---:|
| bob | plp | 0.9 |
| carl | plp | 0.6 |
| greg | plp | 0.7 |
| ian | db | 0.9 |
| harry | db | 0.8 |

Table 1: *author/2*

| researcher | university | P |
|:---:|:---:|:---:|
| edwin | harvard | 1.0 |
| fred | harvard | 0.9 |
| alice | mit | 0.6 |
| dave | mit | 0.7 |

Table 2: *location/2*

| researcher | researcher | P |
|:---:|:---:|:---:|
| alice | edwin | 0.2 |
| alice | fred | 0.3 |
| bob | carl | 0.4 |
| bob | greg | 0.5 |
| bob | harry | 0.6 |
| bob | ian | 0.7 |
| carl | greg | 0.8 |
| carl | harry | 0.9 |
| carl | ian | 0.8 |
| dave | edwin | 0.7 |
| dave | fred | 0.6 |
| edwin | fred | 0.5 |
| greg | harry | 0.4 |
| greg | ian | 0.3 |
| ian | ian | 0.2 |

Table 3: *coauthor/2*

that are located at some university and are coauthors with someone, then the answer consists of the substitutions $[z/\mathsf{edwin}]$, $[z/\mathsf{alice}]$ and $[z/\mathsf{dave}]$.

A *Boolean query* is a query with fully quantified variables. A *conjunctive query* (CQ) is a negation-free first-order logic conjunction with all variables either free or bound by an existential quantifier. A *union of conjunctive queries* (UCQ) is a disjunction of conjunctive queries. We mainly work with UCQs in this paper.

Conjunctive queries may also be expressed using Prolog notation [Clocksin and Mellish, 1981]. For instance, the rule $\mathsf{R}(z) \,{:}{-}\, \mathsf{location}(z, x), \mathsf{coauthor}(z, y).$ represents the query $Q_2$ from Example 1. Here, $\mathsf{R}(z)$ is called *head* of the rule[1] and $\mathsf{location}(z, x), \mathsf{coauthor}(z, y)$ is called the *body*. The rule is understood as defining tuples of a new table $\mathsf{R}$ that, in this case, contains all tuples $[t]$ where $[z/t] \in \mathrm{Ans}(\exists x, y.\ \mathsf{location}(z, x) \wedge \mathsf{coauthor}(z, y), \omega)$. Thus, variables not occurring in the head of the rule correspond to existentially quantified variables in the respective conjunctive query. In Prolog notation, UCQs are represented as a collection of rules with the same relation in the head. For a given set of rules representing a UCQ, we denote *prediction set* as the union of its respective answer sets.

## 2.2 Probabilistic Databases (PDBs)

A database $\mathcal{D}$ is said to be a PDB if its every tuple has an assigned probability value. Mathematically, A PDB $\mathcal{D}$, for a vocabulary $\sigma$, is a finite set of tuples of the form $\langle t, p \rangle$, where $t$ is a $\sigma$-atom and $p \in [0, 1]$. Moreover, if $\langle t, p \rangle \in \mathcal{D}$ and $\langle t, q \rangle \in \mathcal{D}$, then $p = q$. Each PDB for the vocabulary $\sigma$ induces a unique probability distribution over $\sigma$-interpretations $\omega$:

$$P_{\mathcal{D}}(\omega) = \prod_{t \in \omega} P_{\mathcal{D}}(t) \prod_{t \notin \omega} (1 - P_{\mathcal{D}}(t)), \tag{1}$$

---

1. The choice for the relation $\mathsf{R}$ ($\notin \mathcal{R}$) was arbitrary here.

where

$$P_{\mathcal{D}}(t) = \begin{cases} p & \text{if } \langle t, p \rangle \in \mathcal{D} \\ 0 & \text{otherwise.} \end{cases}$$

**Example 2** Tables 1, 2 and 3 form a PDB with 3 relations: coauthor/2, author/2 and location/2.

Furthermore, the probability of any Boolean query $Q$ w.r.t. a PDB $\mathcal{D}$ is

$$P_{\mathcal{D}}(Q) = \sum_{\omega \models Q} P_{\mathcal{D}}(\omega). \tag{2}$$

**Example 3** For the Boolean query $Q_3 = \text{coauthor}(\text{bob}, \text{carl})$ we can read off the probability $P_{\mathcal{D}}(Q_3) = 0.4$ directly from Table 3. On the other hand, the probability of query $Q_4 = \exists x. \, \text{coauthor}(\text{bob}, x) \wedge \text{coauthor}(\text{carl}, x)$ cannot be read directly from any table and requires probabilistic inference. Directly using Equation 2, we get $P_{\mathcal{D}}(Q_4) = 0.879$. Intuitively, $Q_3$ asked about the probability that bob and carl are co-authors, whereas $Q_4$ asked about the probability that bob and carl have a common co-author.

### 2.3 Lifted Inference

A query plan is a sequence of database operations (namely join, projection, union, selection and difference) that are executed to do exact probabilistic inference for a query [Van den Broeck and Suciu, 2017]. Extensional query plans are query plans under the tuple-independence assumption. Extensional query plans can be used to compute probabilities of queries over probabilistic databases. However, not all queries have an extensional query plan that can be used to compute their probabilities correctly. Those for which such a correct extensional query plan exists are called *safe queries*. Hence, safe queries can be evaluated in time polynomial in the size of the database (data complexity). In general, for every UCQ, the complexity of evaluating it on a probabilistic database is either PTIME (safe queries) or #P-complete in the size of the database [Dalvi and Suciu, 2012]. The algorithms with PTIME data complexity are also called *lifted inference algorithms*. In this work, we exploit the lifted inference algorithm of $Lift_R^O$ which is an extension of the $Lift^R$ algorithm [Gribkoff et al., 2014]. The $Lift_R^O$ algorithm exploits the structure of a query and uses a divide and conquer approach to calculate its probability.

### 2.4 Probabilistic Rules

When we have a probabilistic database and a collection of Prolog rules defining a relation $R$, the relation $R$ becomes a random variable.[2] The relation $R$ does no longer have to be representable as a tuple-independent table, though. Additionally, one may enlarge the set of probability distributions over tuples of $R$ that can be modelled by Prolog rules by introducing auxiliary probabilistic tables, not originally present in the database and using them in the rules. In this way, one can think of Prolog rules as a model of a distribution

---

2. Here we use the term "random variable" in a broad sense; similarly as we may have matrix-valued random variables, we may also have random variables that represent database relations.

rather than only as a way to query a pre-existing probabilistic database. This is illustrated by the next example.

**Example 4** As an example, let us consider again the database that is listed in Tables 1, 2 and 3 and a rule $R(x, y) :–$ coauthor$(x, y)$. Suppose that we introduce a new probabilistic relation $\mathsf{A}(x, y)$ with all possible $\sigma$-atoms $\mathsf{A}(t_1, t_2)$ having the same probability $p$ and replace the original rule by $R(x, y) :–$ coauthor$(x, y), \mathsf{A}(x, y)$. By changing the parameter $p$ we can now decrease the probabilities of the $\sigma$-atoms $R(t_1, t_2)$.

We use the following notation as syntactic sugar for probabilistic rules. We write $p :: R(x, y) :–$ coauthor$(x, y)$ to denote $R(x, y) :–$ coauthor$(x, y), \mathsf{A}_{R,i}(x, y)$ where we set $P_{\mathcal{D}}[\mathsf{A}_{R,i}(t_1, t_2)] = p$ for all tuples $(t_1, t_2)$ and where $i$ is an identifier of the respective rule. For rules with variables appearing in the body but not in the head, such as $p :: R(x, y) :–$ author$(x, z),$ author$(y, z)$, the auxiliary relation's arguments range only over the variables in the head; that is for the considered rule we would have $R(x, y) :–$ author$(x, z),$ author$(y, z), \mathsf{A}_{R,i}(x, y)$. This restriction is necessary in order to keep the resulting rules simple enough so that lifted inference could still be used, i.e. we want to avoid the introduction of rule's probabilistic weights to result in creating unsafe queries from safe ones. Moreover, since we will only be querying probability of individual (ground) tuples, it will be possible to create the relation $\mathsf{A}_{R,i}(x, y)$ always on the fly to contain just one tuple.

As mentioned above, relations defined using probabilistic rules do not have to be tuple-independent. Since the main intended application of the present work is to be able to fill in missing probabilistic tuples into tuple-independent probabilistic databases, we introduce the following operation, denoted $\text{Ind}(R)$, which produces a tuple-independent relation $R'$ from a given relation $R$ by simply assigning marginal probabilities to the individual tuples. We may notice that when $R$ is defined using probabilistic rules and deterministic relations, i.e. relations where every tuple has probability either 0 or 1, then $R = \text{Ind}(R)$. This can also be seen as materializing views to tuple-independent tables.

## 3. Problem Specification

We aim to learn probabilistic rules from a given PDB $\mathcal{D}$ and a target relation, $target$. We call $E$ the set of the tuples, of $target$, present in $\mathcal{D}$. We could interpret the set of these probabilistic training examples $E$ as defining a distribution over deterministic training sets in the same way as a probabilistic database represents a distribution over deterministic databases. Then, given a probabilistic database, during training we search for a collection of probabilistic rules so that we would obtain a *good* model for the data.

**Example 5** Consider a set of examples for a relation supervisor: $E = \{\langle[\mathsf{greg}, \mathsf{carl}], 0.8\rangle,$ $\langle[\mathsf{edwin}, \mathsf{fred}], 0.4\rangle\}$ for the database from Tables 1, 2, 3. The task is then to find rules that define the relation supervisor using the relations author, location and coauthor. For instance, one such, not particularly optimal, model could be $0.1 ::$ supervisor$(x, y) :–$ coauthor$(x, y)$.

What we will call a *good* model, depends on the context (in the next section, we consider different loss functions which could lead to different ways to score the learned probabilistic models). Moreover, what is deemed a *good* model also depends on the assumptions about the way the training tuples were obtained and on what we assume about the tuples that

are not present in the set of training examples. We now describe the problem Statement formally as follows.

**Given:** PDB $\mathcal{D}$, a target relation *target*

**Find:** A set of probabilistic rules called $H = \{h_1, h_2, \ldots, h_n\}$ with *target* in the head of each rule such that cross entropy loss (cf. Equation 3) is minimum over all *target* tuples in $\mathcal{D}$.

SafeLearner addresses the above Stated problem heuristically using two main components. 1) A structure learning component that generates candidate rules — a larger set of rules that subsumes $H$. 2) A parameter learning component that optimizes weights of these rules thus effectively also performing selection of these rules as a result. We present each of these components in Sections 4 and 5 respectively before explaining the algorithm of SafeLearner in Section 6.

## 4. Structure Learning

Most relations of KBs have binary relations and we restrict ourselves to it. In order to learn structure in this specified problem setting, we learn deterministic candidate rules. This method of structure learning without considering the parameters $p_H$ is simpler than the case when the structure and parameters are learned jointly. But this method is followed keeping in mind that the parameters are tuned later. We use AMIE+ for generating candidate rules. It is a highly efficient rule generation system for mining deterministic rules from KBs. It uses a language bias (constraints on the form of mined rules) to mine the rules from a vast search space. Compared with a naive support-based confidence score [Agrawal et al., 1993], it uses a more involved scoring function. By *confidence*, we refer to the confidence score used by AMIE+, throughout the rest of the paper. A rule is output when it satisfies some properties [Galárraga et al., 2015], its confidence score surpasses a specified confidence threshold, and improves over the confidence scores of all its parents.

### 4.1 Generating Candidate Rules

Since AMIE+ expects deterministic input, we have to first generate a deterministic database from the given probabilistic database and the given training set. This may be achieved either directly by sampling from the database or by keeping only tuples that have probabilities over some threshold. The rationale behind this simple approach is as follows. Let us suppose that the training data was generated by directly sampling from a model consisting of a probabilistic database together with some probabilistic rules. Suppose that the rules contained the rule $0.7 ::$ author$(a, p)$ :– author$(b, p)$, supervisor$(a, b)$. We may then reasonably expect that a deterministic rule learner, such as AMIE+, will be able to find the respective deterministic rule since its confidence should be on average at least 0.7 on datasets sampled form the data-generating model. At least on a heuristic level, this justifies using a deterministic rule learner for finding candidate rules. Ideally, this intermediate deterministic step wanes the predictive performance in comparison to the case where candidates could be obtained on the probabilistic tuples [De Raedt et al., 2015] but this slight decrease in performance is compensated by the improvement in the scalability provided by AMIE+. Unless specified otherwise by the user, SafeLearner uses AMIE+ restricted to rules with two atoms in the

body. Then, the algorithm selects $k$ rules with highest confidence as the candidates, where $k$ is a parameter of the algorithm, and checks if the resulting rule set correspond to a safe UCQ. If the UCQ is not safe, it removes the minimal number of lowest-confidence candidate rules that make the resulting UCQ safe. Finally, we add a rule for the *target* relation that has an empty body. Whenever this rule has non-zero probabilistic weight, all the possible tuples will have non-zero probability (this turns out to be important for avoiding infinite cross entropy scores).

## 5. Parameter Learning

Since we are learning a distribution over databases, the natural choice is to measure the likelihood of the model consisting of the learned rules given the training data. However, our training datasets are also probabilistic themselves. Thus, we need to measure how 'close' the learned distributions are to the distributions of the training data. This is typically measured by the KL divergence. Another way to look at this problem is by asking what the likelihood would be given the data sampled from the training distribution. This is an expectation of likelihood, or cross entropy, which turns out, is equivalent to KL divergence (up to a constant) when we assume both independence of the tuples and independence of the predictions (i.e. we replace the target relation $R$ by its materialization $R' = \mathrm{Ind}(R)$). This can be seen as follows. First, let us assume that the training examples are complete, that is, the training set $E$ contains all possible tuples $t_i$ and their respective probabilities $p_i$. Then, we can write the expected log-likelihood of the model given the training examples as follows. Let $\mathbf{T}$ be the set of all possible tuples. For every $i \in \{1, \ldots, |\mathbf{T}|\}$, let $T_i$ be a Bernoulli random variable corresponding to the $i$-th training example $\langle t_i, p_i \rangle \in E$ such that $P[T_i = 1] = p_i$. Next let $q_i$ denote the probability assigned by the model to the tuple $t_i$. Then we may write the expected log-likelihood as

$$\mathbb{E}\left[\log\left(\prod_{t_i \in \mathbf{T}} q_i^{\mathbb{1}(T_i=1)} \cdot (1-q_i)^{\mathbb{1}(T_i=0)}\right)\right] = \sum_{\langle t_i, p_i \rangle \in E} (p_i \log q_i + (1-p_i) \log(1-q_i)) \quad (3)$$

where the expectation is over the samples of the probabilistic database (here represented by the variables $T_i$). This is the same as cross entropy, except for the sign. Hence, what we need to minimize is indeed cross entropy.

### 5.1 Estimating Losses from Incomplete Training Data

Naturally, in most realistic cases, access to probabilities of all the possible tuples is limited. We introduce two distinct settings that lead to a different way of treating tuples that are outside the training set.

#### 5.1.1 Learning Models for the Whole Database

In the first setting, we assume that our goal is to learn a probabilistic model for the whole database. Since most domains will be sparse, we may assume that a majority of the tuples (over elements of the given domain) will have some very small probability $\gamma_{target}$, possibly zero. When estimating the loss of a set of rules $Q$ (here represented as a UCQ), we need

to split the possible *target* tuples (Cartesian product of sets of constants of all the entities constituting the *target*) into three categories:

1. *target* tuples contained in the training set, also denoted by the set of examples $E$.

2. possible *target* tuples outside $E$ that lie in the answer of $Q'$ (i.e. in $\text{Ans}(Q')$), where $Q'$ contains all the rules in $Q$ with non-empty bodies. These are all the *target* tuples outside $E$ that have at least one grounded proof in the training data. In other words, these tuples can be inferred from the training data by the rules in $Q'$.

3. the remaining possible *target* tuples. These tuples are neither present in the training data nor can these be inferred from the training data by the rules in $Q'$.

The estimate of the loss is then obtained as the sum of estimates of these three losses. Estimating the first and second component of the loss is straightforward. To speed it up, we estimate the second component only on a sample of the tuples of the second category (as there may often be significantly more tuples in this category than in the first one). Similarly, it is not difficult to estimate the third component of the loss because it only depends on the probability of the rule with empty body and all tuples in this third category have the same estimated probability.

We will denote by $E$, $E_2$, $E_3$ the sets of sampled tuples for categories 1, 2 and 3, respectively. The weights of tuples in $E_2$ and $E_3$ are calculated corresponding to the inverse of the subsampling ratio. For example, if we sample 5000 tuples out of a possible set of 100000, we assign a weight of 20 to each of the sampled tuples. Tuples in $E_2$ are assigned a weight $w_2$ and tuples in $E_3$ are assigned $w_3$.

### 5.1.2 LEARNING MODELS FOR SUBSETS OF DATABASE

In the second setting, which is closer to the setting considered in De Raedt et al. [2015], we assume that our task is not to obtain a good model for the whole database. Instead, we assume that there is a distribution that generates probabilistic tuples for which we, then, need to predict the probability with which these tuples hold. It is important to stress that the distribution that generates the tuples is not the distribution that we need to model. As emphasized by De Raedt et al. [2015], one may also see this setting as a probabilistic extension of the PAC-learning framework [Valiant, 1984], and, in particular, of its probabilistic extension [Kearns and Schapire, 1994]. It emerges that, in this setting, we can ignore tuples that are not in the training set. An instance where this setting is natural is, for example, automatic construction of knowledge bases from text or other non-structured source where we need to learn rules to estimate probability with which the tuples suggested by the construction process really hold. In contrast to the previous setting, we do not use the rules to predict anything about any 'new' tuples.

## 5.2 Probabilistic Inference and Weight Learning

To our best knowledge, Slimshot [Gribkoff and Suciu, 2016] was the first lifted inference engine developed in 2016 that also performed approximate probabilistic inference for unsafe queries using sampling. We used its latest version [Gribkoff, 2017] with the $Lift_R^O$ algorithm and extended it so that it could not only do numeric exact inference for safe queries but

could also do it in a symbolic setting. In our implementation, it returns the probability of a query (constructed from $H$) as a closed-form function of the rule probabilities.

Once we have a set of candidate rules $H$, we need to learn its rule weights $p_H$. For this, we first need to predict probabilities of all *target* tuples with respect to $H$ and compute the loss. We begin this process by converting $H$ into a UCQ $Q$. But, before calling Slimshot for lifted inference on the whole query, we first break down the query to independent sub-queries such that no variable is common in more than one sub-query. Then, we perform inference separately over it and later unify the sub-queries to get the desired result.

### 5.3 Initialization of rule weights ($p_H$)

SafeLearner initializes the probabilistic weights of the rules and sets them equal to their confidences estimated from the training data. The confidence of a rule is an empirical estimate of the conditional probability of the rule's head being true given its body is true. In other words, for any rule $h_i$ in $H$, its probability $p_{h_i}$ is initialized as the classical conditional probability of $\mathrm{P}(head = True | body = True)$.

$$p_{h_i} = \frac{|\text{Prediction set of rule } h_i \cap E|}{|\text{Prediction set of rule } h_i |}$$

Lastly, we perform Stochastic Gradient Descent (SGD) to optimize the rule weights.

## 6. SafeLearner

In this section, we describe the SafeLearner system that tackles the problem Statement defined in the preceding section. As its input, SafeLearner obtains a probabilistic database and a target relation and returns a set of probabilistic rules for the target relation. SafeLearner learns only safe rules for which inference has polynomial-time data complexity. It consists of two main components: a structure learning component that generates candidate rules and a parameter learning component that optimizes weights of these rules (and as a result also effectively performs selection of these rules).

### 6.1 Algorithm

Algorithm 1 outlines the working of SafeLearner[3]. Similar to ProbFOIL, the algorithm takes input a PDB $\mathcal{D}$ and a *target* relation as a Problog file. The input file could also specify the type constraints among the relations, if any. Also, the choice of the loss function (which is cross-entropy by default) could be given as a parameter. Once $\mathcal{D}$ is parsed from the input file, all the tuples are separated with their probabilities and are given as input to AMIE+ (cf. 4.1). In line 3, the deterministic rules that are output by AMIE+ are searched for type consistency and are put together as $H$ — a hypothesis of deterministic candidate rules. Further, in lines 4 - 5, we initialize the probability $p_{h_i}$ (cf. 5.3) for every candidate rule $h_i$. Once the probabilities are embedded in the rules, we now have $H$ as a set of probabilistic candidate rules. To make our learned rules extensible to all the possible *target* tuples, we sample more *target* tuples in line 6 as explained in 5.1.1. In line 7, we formulate a UCQ

---

3. Full paper with appendices, and SafeLearner code is available at https://github.com/arcchitjain/SafeLearner/tree/AKBC19

---

**Algorithm 1** SafeLearner – Main Algorithm

---

1: **Input:** PDB $\mathcal{D}$, *target*
2: $E :=$ Set of all *target* tuples in $\mathcal{D}$
3: $H :=$ Set of all the type consistent and significant (deterministic) rules from AMIE+ using $\mathcal{D}$
   with *target* in head
4: Initialize probability $p_{h_i}$ for each rule $h_i$ in $H$
5: Embed rule probabilities $p_H = \{p_{h_1}, p_{h_2}, \ldots, p_{h_n}\}$ in $H$
6: Sample *target* tuples in $E_2$ and $E_3$ and compute their respective sampling weights $w_2$ and $w_3$
7: $Q :=$ QueryConstructor($H$)              ▷ $Q$ is safe
8: **for** $i$ in range(0, MaxIterations) **do**
9:      Randomly select an example $e$ from all the target examples $E \cup E_2 \cup E_3$
10:      $y :=$ ProbabilityPredictor($Q$, $e$)            ▷ $y$ is a function of $p_H$
11:      **if** $e \in E$ **then**
12:         $x :=$ actual probability of $e$ from $E$
13:         Compute cross-entropy loss for $e$ using $x$, $y$ with sampling weight $:= 1$
14:      **else if** $e \in E_2$ **then**
15:         Compute cross-entropy loss for $e$ using $x := 0$ and $y$ with sampling weight $:= w_2$
16:      **else if** $e \in E_3$ **then**
17:         Compute cross-entropy loss for $e$ using $x := 0$ and $y$ with sampling weight $:= w_3$
18:      Get gradient of cross-entropy loss at current $p_H$
19:      Update $p_H$
20: Remove rules with insignificant rule weights from $H$
21: **return** $H$

---

**Algorithm 2** QueryConstructor – Constructs query from a hypothesis $H$ of rules

---

1: $Q :=$ ""
2: **for** (*target*, *body*) in $H$ **do**            ▷ Read a new rule with *target* as head
3:      **if** $Q =$ "" **then**
4:         $Q' := body$
5:      **else**
6:         $Q' := Q +$ " $\vee$ " $+ body$        ▷ Add *body* as a disjunction to existing $Q$
7:      **try**
8:         $p :=$ SlimShot($Q'$)        ▷ SlimShot returns an error if a query is unsafe
9:         $Q = Q'$           ▷ Update $Q$ to $Q'$ as $Q'$ is a safe query
10:      **catch** Unsafe Query Error     ▷ SlimShot throws this error if input query is unsafe
11:         **continue**        ▷ Don't include this rule's *body* as it makes $Q$ unsafe
12:      **end try**
13: **return** $Q$

---

$Q$ from all the probabilistic candidate rules in $H$ (cf. Algorithm 2) Lines 8 - 19 are the heart of this algorithm which use SGD on the loss function, to learn the rule probabilities $p_H$. In line 10, for every randomly selected example $e$, a closed-form function is generated w.r.t $Q$ that evaluates to the predicted probability of $e$ for every specified values of $p_H$ (cf. Algorithm 3 and Appendix D) . Lastly, in line 20, we prune $H$ by removing any rule from it whose probability has diminished very close to 0 (below a predefined tolerance value).

**System Architecture**    SafeLearner is written in Python 2.7 and it relies on Postgres 9.4 as its underlying database management system. It also requires Java for running AMIE+.

---

**Algorithm 3** ProbabilityPredictor – Predicts probability of a UCQ $Q$ for an example $e$

---

1: **Input:** Query $Q$, Example $e$
2: Instantiate the head variables in $Q$ w.r.t. the constants in $e$
3: **if** Number of disjuncts in $Q = 1$ **then**     $\triangleright$ $Q$ cannot be disintegrated to smaller sub-queries
4:     **return** SlimShot($Q$)                    $\triangleright$ Return the probability of the full query $Q$
5: **else**
6:     $DisjunctList :=$ list of all the disjuncts of $Q$
7:     Merge different disjuncts in $DisjunctList$ using disjunctions if they have a common relation
8:     $prob := 1.0$
9:     **for** $SubQuery$ in $DisjunctList$ **do**
10:        $p =$ SlimShot($SubQuery$)                       $\triangleright$ Predict probability of $SubQuery$ for $e$
11:        $prob = prob * (1 - p)$                 $\triangleright$ Unify the probabilities of all the sub-queries
12:     **return** $1 - prob$

---

## 7. Experiments

We empirically address the following two crucial questions experimentally: 1. How does SafeLearner compare against related baselines? 2. How well does SafeLearner scale-up?

### 7.1 How does SafeLearner compare against related baselines?

**Dataset:**   Just like De Raedt et al. [2015], we use the sports subset of NELL KB.[4] For the 850th iteration, it comprises of 8840 probabilistic tuples for the 8 relations mentioned in Appendix A. For a newer iteration (1110), it contains 14234 tuples for the same 8 relations.

**Experimental Setup:**   We investigate the prediction of $target$ tuples from other relations. This is motivated by the problem of predicting tuples that are missing from the database. Here, we aim to learn a set of rules that could be used to predict $target$ from the other relations. For learning the rules, we select the same iteration of NELL that was used by De Raedt et al. [2015] for ProbFOIL[+] (850). For testing, we use the latest iteration (1110) that was available at the time of this experiment. For the correct validation of models, we removed all the tuples from the test data which were present in the training data.

**Baselines:**   The baselines we compare SafeLearner to are the following:

1. **ProbFOIL[+]:** We use the probabilistic rules learned on the NELL knowledge base that were reported by De Raedt et al. [2015]. We were not able to run ProbFOIL[+] in the same way as SafeLearner because ProbFOIL[+] does not scale to large databases. This is mostly because ProbFOIL[+] does not use lifted inference.[5]

The remaining two baselines use the same sets of deterministic rules obtained using AMIE+:

2. **Deterministic rules:** For this baseline, all rules are used with weights set to 1. The rationale for this baseline is to find out to what extent the probabilistic weights influence the predictions on unseen data. We note that even when the rules are

---

4. NELL KB is available at http://rtw.ml.cmu.edu/rtw
5. On the other hand, this also means that ProbFOIL[+] can in theory find more complex rule sets as it does not have to restrict itself to rule sets that result in safe (i.e. liftable) queries.

deterministic, the other tuples in the database are often still probabilistic. Hence, even with deterministic rules, one needs to rely on probabilistic inference.

3. **Probabilistic rules with confidence scores as probabilistic weights:** Although, in general, the probabilistic weights of rules are *not* equal to the conditional probabilities of the head being true given the body is true (i.e. *confidence scores*), we want to assess how much worse the probabilistic predictions will be if we set the probabilistic weights equal to confidence scores estimated on the training data. This baseline is labeled as 'SafeLearner - without learning'.

**Hyper-parameters:** For our experiments, we set the minimum confidence threshold to $= 10^{-5}$. The sample size of both $E_1$ and $E_2$ were specified to be equal to $E$. SGD was run with $10^4$ iterations with learning rate parameter $= 10^{-5}$.

**Results:** We measured cross entropy (cf. Figure 1) and precision-recall (PR) curves (cf. Figure 2) on the test set. For cross entropy, the best results were consistently obtained by SafeLearner which means that the models obtained by SafeLearner were able to provide best estimates of probabilities of the unseen test-set tuples. This is not surprising given that SafeLearner is the only one of the evaluated baseline methods that optimizes a loss function that is a proper scoring rule. Interestingly, the differences were much smaller for the PR curves. This suggests that, at least on our datasets, obtaining ranking of tuples based on predicted probabilities, but not the actual probabilities, can be quite reliably done already by models that are not very precise when it comes to predicting the actual probabilities.

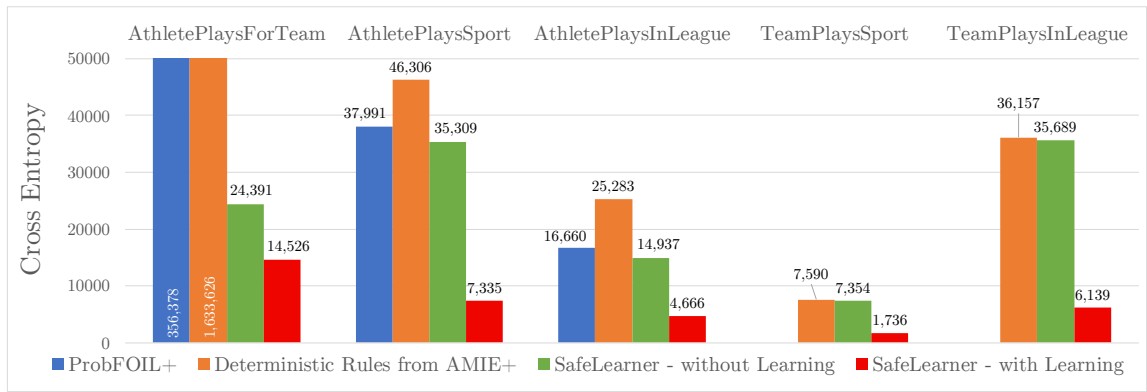

Figure 1: SafeLearner has better cross entropy than its baselines

## 7.2 How well does SafeLearner scale-up?

**Datasets:** This question attempts to test and stretch the limits of SafeLearner by measuring its scalability. We emperically show that SafeLearner scales on the latest iteration of full database of NELL (iteration:1115) with 233,000 tuples and 426 relations. It can also scale up to the standard subset of Yago 2.4 [6] with 948,000 tuples and 33 relations. (For

---

6. Yago is available at http://yago-knowledge.org/

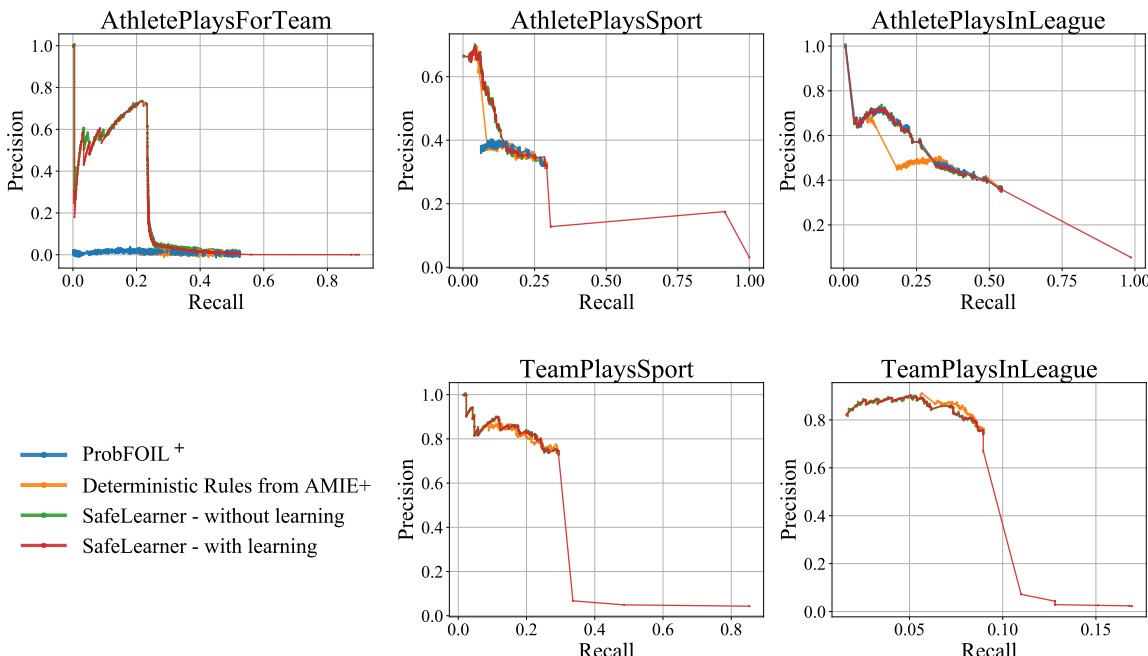

Figure 2: Precision-Recall Curves for 5 relations in NELL Sports Dataset

Yago 2.4, the confidence in the collection of a relation was considered to be the probability of all the tuples of that relation.)

| KB | *target* relation | No. of *target* tuples | No. of rules learned | Runtime |
|------|-------------------|------------------------|----------------------|------------------------|
| NELL | AthletePlaysForTeam | 1687 | 3 | 1 hour  11 minutes |
| NELL | AthletePlaysSport | 1959 | 5 | 2 hours  8 minutes |
| NELL | AthletePlaysInLeague | 1310 | 5 | 1 hour  34 minutes |
| NELL | TeamPlaysSport | 355 | 5 | 5 hours 16 minutes |
| NELL | TeamPlaysInLeague | 1354 | 5 | 3 hours  1 minute |
| YAGO | IsCitizenOf | 14554 | 4 | 2 hours 25 minutes |

Table 4: SafeLearner is able to scale-up to large scale KBs

**Results:**  From Table 4, it can be easily seen that although SafeLearner suffers in runtime when it learns more rules, it is able to scale to more than 14000 *target* tuples (in case of Yago:IsCitizenOf) in relatively lesser time as it learns fewer rules. It would also depend upon how complex the learned rules are. The actual rules learned by SafeLearner are mentioned in the Appendix B. To resolve this trade-off between runtime and number of rules, we could also specify the maximum number of rules to learn as part of the input parameters.

## 8. Related Work and Discussion

The work presented in this paper advances the works [De Raedt et al., 2015, Dylla and Theobald, 2016] that also studied learning in the probabilistic database setting.[7] But compared with these previous works, we rely on lifted inference, which allows our approach to scale to much larger databases. Both of the previous approaches only use tuples from a given training set but do not take into account the behavior of the model on tuples not in the training set. This is problematic because, unless the training set is really large, these previous methods do not distinguish models that predict too many false positives (i.e. models that give too high probability to too many tuples outside the training set). This becomes an issue especially in sparse domains (and most real domains are indeed sparse).

Our work is also closely related to the literature on learning from knowledge bases such as NELL within statistical relational learning (SRL), including works that use Markov logic networks [Schoenmackers et al., 2010], Bayesian logic programs [Raghavan et al., 2012] and stochastic logic programs [Lao et al., 2011, Wang et al., 2014]. A disadvantage of many of these methods is that the learned parameters of the models can not be interpreted easily, which is particularly an issue for Markov logic networks where the weight of a rule cannot be understood in isolation from the rest of the rules. In contrast, the learned weights of probabilistic rules in our work, and also in the other works relying on probabilistic databases [De Raedt et al., 2015, Dylla and Theobald, 2016], have a clear probabilistic interpretation.

**Parameter Learning with Different Losses**   Cross entropy is not the only loss function that may be considered for learning the parameters of probabilistic rules. Here, we discuss two additional loss functions that have already been used for the same or similar tasks in the literature: *squared loss* [Dylla and Theobald, 2016] and a probabilistic extension of accuracy [De Raedt et al., 2015]. Whereas cross entropy and squared loss belong among so-called *proper scoring rules* [Gneiting and Raftery, 2007] and, thus, reward estimates of probabilities that match the true probability, this is not the case for probabilistic accuracy. Moreover, each of these functions also relies on additional assumptions such as mutual independence of the examples' probabilities as well as mutual independence of the predictions, although this is not mentioned explicitly in the respective works [Dylla and Theobald, 2016, De Raedt et al., 2015]. Below, we briefly discuss squared loss and probabilistic accuracy.

**Squared Loss (Brier Score)**   As before, let $p_i$ denote the probability of the $i$-th example and $q_i$ the probability of the respective prediction. Then, the squared loss, which is a proper scoring rule, is: $L_{sq} = \frac{1}{|E|} \sum_{\langle t_i, p_i \rangle \in E} p_i (1 - q_i)^2 + (1 - p_i) q_i^2$. Minimizing $L_{sq}$ is also equivalent to minimizing the loss $L'_{sq} = \frac{1}{E} \sum_{\langle t_i, p_i \rangle \in E} (p_i - q_i)^2$ which was among others used in [Dylla and Theobald, 2016] for learning probabilities of tuples in PDBs.

**Probabilistic Accuracy**   De Raedt et al. (2015) define probabilistic extension of accuracy and other measures of predictive quality such as precision and recall. Their version of probabilistic accuracy is $Acc_{prob} = 1 - \frac{1}{|E|} \sum_{\langle t_i, p_i \rangle \in E} |p_i - q_i|$. Unlike the other two discussed loss functions, probabilistic accuracy is not a proper scoring rule as the next example illus-

---

7. Strictly speaking, the work [De Raedt et al., 2015] was framed within the probabilistic logic programming setting. However, probabilistic logic programming systems, such as Problog [Fierens et al., 2015], can be seen as generalizations of probabilistic databases.

trates, however, it has other convenient properties (cf. De Raedt et al. [2015] for details).

**Example 6** Let the set of domain elements in the database be $\mathcal{C} = \{\mathsf{alice}, \mathsf{bob}, \mathsf{eve}\}$. Next, let $E = \{\langle[\mathsf{alice}], 0\rangle, \langle[\mathsf{bob}], 1\rangle, \langle[\mathsf{eve}], 1\rangle\}$ be the set of training examples for the relation $\mathsf{smokes}$ and let us have the rule $p :: \mathsf{smokes}(X) :\!\!- \mathsf{true}$. Then, maximizing probabilistic accuracy on $E$ yields $p = 1$, whereas optimizing both cross entropy and squared loss would yield $p = 2/3$, which is the probability that a randomly selected person in $\mathcal{C}$ smokes.

## 9. Conclusions

We proposed a probabilistic rule learning system, named SafeLearner, that supports lifted inference. It first performs structure learning by mining independent deterministic candidate rules using AMIE+ and later executes joint parameter learning over all the rule probabilities. SafeLearner extends ProbFOIL$^+$ by using lifted probabilistic inference (instead of using grounding). Therefore, it scales better than ProbFOIL$^+$. In comparison with AMIE+, it is able to jointly learn probabilistic rules over a probabilistic KB unlike AMIE+ which only learns independent deterministic rules (with confidences) over a deterministic KB. We experimentally show that SafeLearner scales as good as AMIE+ when learning simple rules. Trying to learn complex rules leads to unsafe queries which are not suitable for lifted inference. But lifted inference helps SafeLearner in outperforming ProbFOIL$^+$ which does not scale to NELL Sports Database without the help of a declarative bias.

A few limitations of SafeLearner are as follows: 1) It cannot learn complex rules that translate to an unsafe query. 2) It cannot use rules within the background theory. 3) It cannot learn rules on $PDB$ with numeric data (without assuming them as discrete constants).

The main contributions of SafeLearner are presented as follows. Firstly, it accomplishes probabilistic rule learning using a novel inference setting as it is the first approach that uses lifted inference for KB completion. Secondly, unlike ProbFOIL$^+$, SafeLearner scales well on the full database of NELL with 233,000 tuples and 426 relations as well as on the standard subset of Yago 2.4 with 948,000 tuples and 33 relations. Thirdly, SafeLearner is faster than ProbFOIL$^+$ because of the following three factors: 1) it disintegrates longer complex queries to smaller simpler ones, 2) it caches the structure of queries before doing inference and 3) it uses lifted inference to infer on those simple queries. The first two factors of query disintegration and memoization are discussed in Appendix D in further detail.

In future, this work could be advanced further to eliminate its shortcomings. In particular, a prominent direction of advancement would be to extend probabilistic rule learning to open-world setting of which the $Lift_R^O$ algorithm [Ceylan et al., 2016] is capable.

## Acknowledgements

The authors express their sincere regards to Anton Dries and Sebastijan Dumančić for their invaluable input, and the reviewers for their useful suggestions. This work has received funding from the European Research Council under the European Union's Horizon 2020 research and innovation programme (#694980 SYNTH: Synthesising Inductive Data Models), from the various grants of Research Foundation - Flanders, NSF grants #IIS-1657613, #IIS-1633857, #CCF-1837129, NEC Research, and a gift from Intel.

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
