# OpenReview forum: "Scalable Rule Learning in Probabilistic Knowledge Bases"
_AKBC.ws/2019/Conference — AKBC 2019_

### Official Review · AnonReviewer3 · 2019-01-07

**Rating:** 6
**Confidence:** 1

**Review:**

This paper was interesting and rather clearly written, as someone who didn't have much background in rule learning.

Section 5.1.1.1 : "(ii) tuples contained in the answer of Q' where Q' is the same as Q but without the rule with empty body, but not in the training set"  is unclear.

In the algorithm, line 22, aren't rules removed until H leads to a "safe" UCQ ?

Section 6.1 : "In line 7 we formulate a UCQ Q from all the candidate rules in H (explained in 5.2 with an example)". I was unable to find the example in section 5.2

It would be interesting to have an idea of the maximum scale that ProbFoil+ can handle, since it seems to be the only competitor to the suggested method.

In section 7.2 does the "learning time" include the call to AMIE+ ? if not, it would be interesting to break down the time into its deterministic and learning components, since the former is only necessary to retrieve the correct probabilities.

Being new to this subject, I found the paper to be somewhat clear. However, I found that it was sometimes hard to understand what was a part of the proposed system, and what was done in Amie+ or Slimshot.

For example, "But, before calling Slimshot
for lifted inference on the whole query, we first break down the query to independent subqueries such that no variable is common in more than one sub-query. Then, we perform
inference separately over it and later unify the sub-queries to get the desired result."

This is described as important to the speedup over ProbFoil+ in the conclusion, yet doesn't appear in Algorithm 1.

Similarly, "it caches the structure of queries before doing inference " is mentioned in the conclusion but I couldn't map it to anything in Algorithm 1 or in the paper.


I lean toward an accept because the work seems solid, but I feel like I don't have the background required to judge on the contributions of this paper, which seems to me like a good use of Amie+/Slimshot with a reasonable addition of SGD to learn rule weights. Some of the components which are sold as important for the speed-up in the conclusion aren't clear enough in the main text. Some numbers to experimentally back-up how important these additions are to the algorithms would be welcome.

---

> ### Author Response · Authors · 2019-02-01
> **Clarifications addressed**
>
> We would like to thank the reviewer for the comments and would like to clarify further.
>
> 1) Section 5.1.1.1 : "(ii) tuples contained in the answer of Q' where Q' is the same as Q but without the rule with an empty body, but not in the training set" is unclear.
>
> We have elaborated more on the 3 categories of target tuples in the revised paper. We hope it is clearer now.
>
>
> 2) In the algorithm, line 22, aren't rules removed until H leads to a "safe" UCQ?
>
> We have elaborated further on our algorithm in the paper. Checking for a safe UCQ is performed in Line 8 of QueryConstructor function. The only way to check for a safe query is by trying to construct a query plan, which is exactly what happens inside SlimShot. So if SlimShot fails to construct a query plan, the UCQ is considered to be unsafe.
>
>
> 3) Section 6.1: "In line 7 we formulate a UCQ Q from all the candidate rules in H (explained in 5.2 with an example)". I was unable to find the example in section 5.2.
>
> Thank you for pointing it out. It has now been rectified in the revision.
>
>
> 4) It would be interesting to have an idea of the maximum scale that ProbFoil+ can handle since it seems to be the only competitor to the suggested method.
>
> We have conducted a small experiment to demonstrate that ProbFOIL+ struggles to scale upto large KBs with a number of target tuples > 5000. On the other hand, such large KBs are handled reasonably faster by SafeLearner.  For instance, for a simple probabilistic KB with 20000 target tuples and 20000 non-target tuples, ProbFOIL+ took 15 hours and 54 minutes and SafeLearner took just 30 minutes. The detailed procedure and results of the experiment could be found in Appendix C.
>
>
> 5) In section 7.2 does the "learning time" include the call to AMIE+?
>
> Yes, the learning time includes both structure learning (including AMIE+) and parameter learning components.
>
>
> 6) I found that it was sometimes hard to understand what was a part of the proposed system, and what was done in AMIE+ or Slimshot.
>
> Only line 3 in Algorithm 1 uses AMIE+. On the other hand, SlimShot is used in lines 7 (QueryConstructor function) and 10 (ProbabilityPredictor function). Every other line of Algorithm 1 is part of SafeLearner.
>
>
> 7) Explain the breaking down of queries into independent sub-queries in the algorithm.
>
> If a query can be written as a union of independent sub-queries then its probability can be expressed as a unification of the probability of its sub-queries. We have explained this further in Appendix D.
>
>
> 8) Caching is not mentioned anywhere in the algorithm.
>
> We use caching/memoization before calling SlimShot in SafeLearner.  This is primarily done to speed up SafeLearner by storing the results of the expensive function call to SlimShot and returning the cached result when the structure of the query occurs again in the input. Since SlimShot produces a query plan, we exploit the fact that isomorphic queries naturally have the same query plans. We have explained this further in Appendix D.
>
>
> 9) Some of the components which are sold as important for the speed-up in the conclusion aren't clear enough in the main text. Some numbers to experimentally back-up how important these additions are to the algorithms would be welcome.
>
> We performed an experiment where we compared SafeLearner with and without the 2 speed-up techniques, memoization, and query disintegration. Our results on NELL (850th iteration) and YAGO show that memoization and query disintegration give an average speed-up of 50% and 7% respectively. It is important to understand that query disintegration would only give a speed-up when SafeLearner learns a high number of rules with a lot of rules being independent to one another. The detailed procedure and results of the experiment could be found in Appendix D.

---

### Official Review · AnonReviewer1 · 2019-01-08
**Good but not surprising**

**Rating:** 6
**Confidence:** 1

**Review:**

In general, the paper presents a routing practice, that is, apply lifted probabilistic inference to rule learning over probabilistic KBs, such that the scalability of the system is enhanced but being applicable to a limited scope of rules only. I would not vote for reject if other reviewers agree to acceptance.

Specifically, the proposed algorithm SafeLearner extends ProbFOIL+ by using lifted probabilistic inference (instead of using grounding), which first applies AMIE+ to find candidate deterministic rules, and then jointly learns probabilities of the rules using lifted inference.

The paper is structured well, and most part of the paper is easy to follow.

I have two major concerns with the motivation. It reads that there are two challenges associated with rule learning from probabilistic KBs, i.e., sparse and probabilistic nature.
1) While two challenges are identified by the authors, but the paper deals with the latter issue only? How does sparsity affect the algorithm design?

2) The paper can be better motivated, although there is one piece of existing work for learning probabilistic rules from KBs (De Raedt et al. [2015]). Somehow, I am not convinced by the potential application of the methods; that is, after generating the probabilistic rules, how can I apply the probabilistic rules? It will be appreciated if the authors can present some examples of the use of probabilistic rules. Moreover, if it is mainly to complete probabilistic KBs, how does this probabilistic logics based approach compare against embedding based approach?

---

> ### Author Response · Authors · 2019-02-01
> **Kindly refer to the Appendix**
>
> We would like to thank the reviewer for the comments.
>
> 1) The paper identifies sparsity of Knowledge bases as a challenge but does not deal with it. How does sparsity affect the algorithm design?
>
> Since SafeLearner is a Statistical Relational Learning (SRL) approach that would learn first-order rules to predict tuples, it can even reason about new constants being included in the KB without re-training. On the other hand, since knowledge graph embedding (KGE) based approaches implicitly consider all tuples, these may discard the existence of a tuple with a new constant as the embedding for the new constant does not exist. Thus, it is easier for SRL based approaches to handle highly sparse KBs as they can handle the high number of constants since they only learn first-order rules.
>
>
> 2) I am not convinced by the potential application of the methods. After generating the probabilistic rules, how can I apply them? It will be appreciated if the authors can present some examples of the use of the probabilistic rules. If it is mainly to complete probabilistic KBs, how does this probabilistic logics based approach compare against embedding based approach?
>
> We have answered this question in Appendix E. Please have a look.

---

### Official Review · AnonReviewer2 · 2019-01-10
**Sound approach for rule learning but heavy dependence on black-box algorithm to propose candidate rules**

**Rating:** 4
**Confidence:** 4

**Review:**

The paper proposes a model for probabilistic rule learning to automate the completion of probabilistic databases. The proposed method uses lifted inference which helps in computational efficiency given that non-lifted inference in rules containing ungrounded variables can be extremely computationally expensive.
The databases used contain binary relations and the probabilistic rules that are learned, are also learned for discovering new binary relations. The use of lifted inference restricts the proposed model to only discover rules that are a union of conjunctive queries.
The proposed approach uses AMIE+, a method to generate deterministic rules, to generate a set of candidate rules for which probabilistic weights are then learned. The model initializes the rule probabilities as the confidence scores estimated from the conditional probability of the head being true given that the body is true, and then uses a maximum likelihood estimate of the training data to learn the rule probabilities.

The paper presents empirical comparison to deterministic rules and ProbFOIL+ on the NELL knowledge base. The proposed approach marginally performs better than deterministic rule learning.

The approach proposed is straightforward and depends heavily on the candidate rules produced by the AMIE+ algorithm. The paper does not provide insights into the drawbacks of using AMIE+, the kind of rules that will be hard for AMIE+ to propose, how can the proposed method be improved to learn rules beyond the candidate rules.

‘End-to-end differentiable proving’ from NeurIPS (NIPS) 2017 also tackles the same problem and it would be nice to see comparison to that work.

---

> ### Author Response · Authors · 2019-02-01
> **Comparisons to Neural Theorem Provers explained**
>
> 5) ‘End-to-end differentiable proving’ from NeurIPS (NIPS) 2017 also tackles the same problem and it would be nice to see a comparison to that work.
>
> We have qualitatively drawn parallels between our Statistical Relational Learning (SRL) based approach and Knowledge Graph Embedding (KGE) based approaches for the problem of KB completion in Appendix E that we added to the revision. The SRL based approach is much more interpretable and explainable as compared to the black-box KGE based approaches but also, in our opinion, as compared to Neural Theorem Provers (NTPs). KGE based approaches, including NTPs, also need the test data to get the embedding which is not required by any SRL based approaches. SRL based approaches can reason with unseen constants in the data as they learn first-order rules. KGE based approaches would require re-training in order to incorporate a lot of new constants which is not the case with our SRL based approach. Please refer to Appendix E for further details.
>
> To compare the scalability of SafeLearner with NTPs, ‘End-to-end differentiable proving’ uses 4 deterministic KBs that are not at a large scale (Countries KB has 1158 facts, Kinship KB has 10686 facts, Nations KB has 2565 facts and UMLS KB has 6529 facts). Recently, they submitted a follow-up paper, ’Towards Neural Theorem Proving at Scale’, where they claim to have made their technique more scalable. The follow-up paper uses the following 3 deterministic KBs: 1) WordNet18 KB with 151,442 facts, 2) WordNet18RR with 93,003 facts, and 3) Freebase FB15k-237 KB with 14,951 facts. On the other hand, SafeLearner has demonstrated that it scales to YAGO 2.4 KB of 948,000 probabilistic tuples. Although previous SRL techniques were not as scalable, SafeLearner is as scalable as the latest version of NTPs because it is the first SRL technique to use lifted inference.
>
> Moreover, in our opinion, NTPs do not actually 'learn' rules. They enumerate all possible rules up to a defined length and learn how to activate them.  Essentially, NTPs optimize theorem-proving procedure given the rules. In their follow-up paper, they focus on finding just one proof efficiently (instead of all proofs, as in the initial version) and this brings them scalability.

---

> ### Author Response · Authors · 2019-02-01
> **Details on AMIE+ explained**
>
> We would like to thank the reviewer for the comments.
>
> 1) The proposed approach marginally performs better than deterministic rule learning.
>
> This is true only for PR curves but not for cross entropy. In the section of Parameter Learning, we explain how maximizing expected log likelihood is equivalent to minimizing cross entropy (Equation 3). As our proposed approach optimizes on cross entropy, it induces an average reduction in the cross entropy of 82% and 85% as compared to ProbFOIL+ and AMIE+ respectively. The insignificant differences in precision-recall curves only suggest that obtaining a ranking of tuples based on predicted probabilities (but not the actual probabilities) can be quite reliably done already by models that are not very precise when it comes to predicting the actual probabilities.
>
>
> 2) The approach is straightforward and depends heavily on the candidate rules produced by AMIE+.
>
> Our approach is straightforward on purpose as we want to see how much the proper treatment of probabilities in the KB completion task using a rule-based approach helps. At first glance, SafeLearner does seem to be heavily dependent on the rules generated by AMIE+. But AMIE+ is not a black-box as we exactly know the kind of rules we require as candidates. Had AMIE+ not been developed, we could have coded the function ourselves to generate candidate rules. Since AMIE+ is a multi-threaded package in Java, it does the job well by scaling well to large KBs. Furthermore, as SafeLearner is not specific to any particular candidate generation method, it can be used with any other relational rule learner instead of AMIE+.
>
>
> 3) The paper does not provide insights into the drawbacks of using AMIE+, the kind of rules that will be hard for AMIE+ to produce.
>
> As mentioned in the cited paper ‘Fast rule mining in ontological knowledge bases with AMIE+’ (Galarraga et al.,2015), AMIE+ uses 3 types of language biases to restrict the size of the search space:
> 	1) The rules learned by AMIE+ omit reflexive atoms of the form ‘x(A, A)’.
> 	2) The rules are connected, i.e., every atom shares at least one variable transitively to every other atom of the rule. This omits vague rules of the form ‘x(A, B) :- y(C, D)’.
> 	3) The rules are closed, i.e., all the variables in a rule appear at least twice within itself. This omits open rules of the form ‘x(A, B) :- b(A, C)’ which would hold for any substitution of B and C.
>
> Moreover, AMIE+ only works with binary relations in a KB and does not have negations in its rules. For instance, the types of non-recursive rules of length <= 3, that can be generated by AMIE+ within SafeLearner are:
> 	  1) x(A, B) :- y(A, B).
>     	  2) x(A, B) :- y(B, A).
>
>     	  3) x(A, B) :- y(A, C), y(C, B).
>     	  4) x(A, B) :- y(A, C), y(B, C).
>     	  5) x(A, B) :- y(C, A), y(C, B).
>     	  6) x(A, B) :- y(C, A), y(B, C).
>
>     	  7) x(A, B) :- y(A, C), z(C, B).
>     	  8) x(A, B) :- y(A, C), z(B, C).
>     	  9) x(A, B) :- y(C, A), z(C, B).
>      	10) x(A, B) :- y(C, A), z(B, C).
> In the context of doing Probabilistic Rule Learning for KB Completion, these are precisely all the forms of rules which we require as we can not practically predict missing tuples using reflexive, disconnected or open rules. SafeLearner is capable of learning rules of any length by specifying the maximum rule length as an input parameter.
>
>
> 4) How can the proposed method be improved to learn rules beyond the candidate rules?
>
> The method can be used with any other method that learns deterministic rules to make them probabilistic. We do not claim that using AIME+ is the best. Our main interest is in seeing if/how the proper treatment of probabilities helps in the KB completion tasks using a rule-based approach.

---

### Meta-Review · Area_Chair1 · 2019-02-09
**Nice paper but concerns about related work need to be addressed**

**Recommendation:** Accept (Poster)
**Confidence:** 4

**Metareview:**

The paper presents a method of learning probabilistic rules from
a probabilistic dataset of KB tuples.  They first use existing
deterministic rule-learning algorithm AMIE+ to get candidate
rules and then learn probabilistic rules using lifted inference.
The paper is written clearly.  Authors have responded the
reviewers' concerns well.  Overall there are some concerns that the
contributions of the paper are not substantial enough in quantity
and depth.  Given the vast existing literature on the topic,
the authors should try to resolve the questions of comparisons
that naturally arise.

---

### Decision · Program_Chairs · 2019-02-15
**AKBC 2019 Conference Decision**

Accept